# Quantifying the carbon footprint of clinical trials: guidance development and case studies

Jessica Griffiths ![ORCID],[1] Lisa Fox,[1] Paula R Williamson,[2] on behalf of the Low Carbon Clinical Trials Group

[1]The Institute of Cancer Research, London, UK
[2]The University of Liverpool, Liverpool, UK

**Correspondence to**
Jessica Griffiths;
jessica.griffiths@icr.ac.uk

## ABSTRACT

**Background** The urgency of the climate crisis requires attention from biomedical research, not least clinical trials which can involve significant greenhouse gas emissions. The Low Carbon Clinical Trials Working Group set out a strategy to reduce the emissions of clinical trials, starting with the development of a method to measure their carbon footprint ($CO_2$e).

**Methods** As a first step, we developed a process map defining clinical trial core activities. Corresponding emission factors were sourced to convert activity data into greenhouse gas emissions. The subsequent method was applied to two Cancer Research UK (CRUK)-funded trials (the international randomised sarcoma trial CASPS (ISRCTN63733470) and the UK cohort-based breast cancer trial PRIMETIME (ISRCTN41579286)). A guidance document defining the scope, method and assumptions was written to allow application to any publicly funded/investigator initiated clinical trial.

**Results** Trial specific activities over and above routine care were grouped into 10 modules covering trial set up, conduct and closure. We identified emission factors for all trial activities within both trials and used them to estimate their total carbon footprint. The carbon footprint of CASPS, an international phase 2 trial of an investigational medicinal product with 47 participants, was 72 tonnes $CO_2$e, largely attributable to clinical trials unit emissions and staff travel. PRIMETIME, a UK-based phase 3 non-investigational medicinal product trial with 1962 patients, produced 89 tonnes $CO_2$e, largely attributable to trial-specific in-person participant assessments.

**Conclusion** We have developed a method and guidance that trialists can use to determine the carbon footprint of clinical trials. The guidance can be used to identify carbon hotspots where alternative approaches to trial design and conduct could reduce a trial footprint, and where methodology research is required to investigate the potential impact of interventions taken to reduce carbon emissions. We will continue to refine the guidance to increase the potential application and improve usability.

## STRENGTHS AND LIMITATIONS OF THIS STUDY

⇒ This is the first published and publicly available method and guidance to carbon footprint a clinical trial that can be used to inform lower carbon trial design.
⇒ The guidance has been developed by an extensive and broad reaching collaboration and is intended for use by trialists with no prior experience of carbon footprinting.
⇒ The method and guidance accounts for the carbon footprint of a clinical trial but no other metrics that may be important to consider in a full life cycle analysis.
⇒ The emission factors used are the most appropriate publicly available factors that we could identify. More suitable or up to date emission factors may be available in the future or via paid subscription.
⇒ The validity of results generated by the guidance is dependent on the quality and completeness of the activity data collated and the emission factors used.

## INTRODUCTION

The WHO has called climate change 'the single biggest health threat facing humanity today'.[1] While clinical trials are critical to identifying effective and safe treatments, in line with all healthcare activities, they also have a significant environmental impact. This contribution was first recognised around 16 years ago, when Ian Roberts and other members of the Sustainable Trials Study Group concluded that 'clinical trials contribute substantially to greenhouse gas emissions…. Notably through energy use in research premises and air travel'.[2] Another study conducted by Lyle *et al* of 12 UK pragmatic randomised trials, involving an average of 402 participants, showed that the average carbon emissions generated by the trials was 78.4 tonnes.[3] Multiplying this total by the 350 000 national and international trials registered on ClinicalTrials.gov would estimate the emissions of all global trials to be about 27·5 million tonnes of carbon dioxide equivalent.[4] This total, which is likely a highly conservative estimate, is comparable to the 25 million tonne $CO_2$e total footprint of the UK National Health Service (NHS) which accounts for 6% of the UKs total footprint.[5 6] More recently, Mackillop *et al* published a study quantifying

the carbon footprint of three industry-sponsored late-stage cardiovascular, oncology and respiratory clinical trials.[7] The total carbon footprint of the trials was 2498 tonnes $CO_2e$, 1632 tonnes $CO_2e$ and 1437 tonnes $CO_2e$, respectively.

However, with the exception of the aforementioned industry-sponsored study by Mackillop *et al*, little seems to have happened to quantify and consciously reduce the carbon consumption of clinical trials, but the urgency of the threat from climate change has increased exponentially.

It is not currently possible for trialists to easily estimate the environmental impact of running a clinical trial which limits their ability to contribute to climate change mitigation. However, there is recognition of the requirement to reduce the carbon footprint of clinical trials and practical guidance available encouraging this.[8]

The Sustainable Healthcare Coalition (SHC) brought together the Low Carbon Clinical Trials (LCCT) Working Group including members of the MRC-NIHR Trials Methodology Research Partnership (TMRP), UK and Ireland trialists led by the Institute of Cancer Research Clinical Trials and Statistics Unit (ICR-CTSU) and the Liverpool Clinical Trials Centre, clinicians and others, to set out a strategy to reduce the carbon footprint of clinical trials.[4] The first step to reducing the carbon footprint of a planned clinical trial is to reliably measure its potential footprint and identify the carbon hotspots. Subsequently, trialists will be able to interrogate alternative trial design approaches that may reduce the carbon footprint without impacting data quality, integrity and validity.

Through an NIHR funded project (award ID: NIHR135419), we have developed a method and associated guidance that can be used by trialists to carbon footprint a clinical trial to inform future lower carbon trial design.[9] The aim is for the guidance to be applied prospectively during the design phase of a trial before trial funding is secured. In this paper, we explain how the project team developed the guidance; illustrate initial results from its application on two Cancer Research UK (CRUK) funded trials managed by ICR-CTSU and describe the ongoing work to refine and expand the guidance.

## METHODS

The approach to estimating the carbon impact of a publicly funded and/or investigator initiated clinical trial comprised the following stages:

1. The development of a process map to capture all activities of a clinical trial.
2. The development of a method to quantify the carbon footprint of all trial activities conducted specifically for the trial and which are over and above the provision of routine care, as defined in the process map.
3. Testing of the methodology with selected clinical trials to identify activity carbon hotspots and opportunities for mitigation.

### Process mapping

In order to calculate the carbon footprint of any product, process or service, the whole system must be understood and established. To do this, information was extracted from available sources such as the ICR-CTSU Quality Management System and the NIHR Clinical Trials Toolkit. In addition, a series of information gathering meetings were held with participating clinical trials units (CTUs) (table 1) to understand their trial portfolios and identify clinical trials and activities not yet represented in the process map. Discussions resulted in refinement of the process map and identification of a trial within each CTU to footprint which would result in the inclusion of the widest variety of trial types and activities in the test phase. The output of this stage was a visual representation of a clinical trial and a list of activities considered to be core clinical trial activities, as shown in figure 1.

### Guidance development

The core trial activities were grouped into the following modules: Trial set up; CTU emissions (includes energy and heating used in research premises, trial staff commuting and statistical analysis); Trial specific meetings and travel; Treatment intervention; Data collection and exchange; Trial supplies and equipment; Trial specific patient assessments; Samples; Laboratory and Trial close out.

The carbon footprint of a trial was calculated by multiplying activity data by emission factors. Only activities undertaken to answer the research question, over and above routine care, were included. The main emission factor sources used were Ecoinvent V.2.2, GOV.UK GHG conversion factors and the SHC care pathway carbon calculator.[10 11] Where factors were not available in these publicly available sources, peer-reviewed, published papers, articles and product specifications (eg, for pathology tests and electronic devices) were used. The most applicable and up to date factors that are freely available for public use were selected, and all sources cited and referenced. It is important to note that in some cases more up to date factors, or forecasted emission factors may be available, however, we wanted to ensure that the guidance developed could be used without the need for purchasing licenses for those various emission factors, which would be a barrier for publicly funded trialists.

Two documents were produced, a 'Detailed Guidance and method to calculate the carbon footprint of a clinical trial' and a 'Data collation quick guide and worksheet'. The detailed guidance defines the project scope, limitations, assumptions, emission factors and benchmark data sources, as well as detailed breakdowns of the calculations required to calculate the carbon footprint of each activity. The 'Data collation quick guide and worksheet' guides users in the application of the detailed guidance by summarising the calculations in their simplest form and displaying the activity data the user is required to collect in order to complete the required calculations. See online supplemental appendices A and B for the current version of the detailed guidance and worksheet, respectively.

**Table 1** Participating CTUs testing the guidance and trials undergoing footprinting

| CTU | Trial name | Description |
| --- | --- | --- |
| Cardiff Centre for Trials Research | The UK stand together trial | A two-arm pragmatic multicentre cluster randomised controlled trial which aims to evaluate the effectiveness and cost-effectiveness of KiVa, a school-based anti-bullying programme, in reducing bullying in schools compared with usual practice. 116 primary schools participated from four areas: North Wales, West Midlands, South East and South West England.[17] |
| Edinburgh Clinical Trials Unit | RESTART | A prospective, open, blinded end point, parallel group randomised clinical trial that compared the effects of starting vs avoiding antiplatelet therapy after intracerebral haemorrhage. The trial recruited 537 participants at 122 hospitals in the UK.[18] |
| Imperial Clinical Trials Unit | ON-PACE | On-PACE is a double-blind randomised trial investigating whether taking a nutritional supplement is beneficial for people with the most severe form of chronic obstructive pulmonary disease (COPD). It will recruit 96 people with COPD who use oxygen at home to take part in a 3-month long clinical trial.[19] |
| Liverpool Clinical Trials Centre | HEAL-COVID | HElping Alleviate the Longer-term Consequences of COVID-19 (HEAL-COVID), an adaptive platform trial, aims to evaluate the impact of treatments on longer-term morbidity, mortality, re-hospitalisation, symptom burden and quality of life associated with COVID-19. The trial took place across 109 sites and randomised 1245 participants.[20] |
| MRC Clinical Trials Unit at UCL | MAVMET | A multicentre, 48-week randomised controlled factorial trial of adding maraviroc and/or metformin for hepatic steatosis in HIV-1-infected adults on combination antiretroviral therapy. The trial took place at six sites across the UK and recruited 90 participants.[21] |
| Newcastle Clinical Trials Unit | PREMISE | A multi-arm, multicentre, non-inferiority randomised controlled trial comparing three minimally invasive treatments to the current gold standard operation for bladder obstruction due to enlarged prostate in the National Health Service. The planned sample size is 536.[22] |
| The George Institute | INTERACT3 | An international, multicentre, prospective, stepped wedge, cluster randomised, blinded outcome assessed, controlled trial of a care bundle of physiological control strategies in acute intracerebral haemorrhage. The trial recruited 7064 patients from 122 hospitals in 10 countries (Chile, Brazil, China, India, Mexico, Nigeria, Pakistan, Peru, Sri Lanka and Vietnam).[23] |
| University of Galway | EMERGE | A randomised placebo-controlled trial of the Effectiveness of MEtformin in addition to usual care in the Reduction of GEstational diabetes mellitus effects. Planned sample size is 550.[24] |

CTUs, clinical trials units.

### Applying the guidance—case studies

Although we intend for the guidance to be used prospectively at the trial design stage, in order to test the method and develop a full and comprehensive set of calculations, this guidance was retrospectively and iteratively applied to two ICR-CTSU managed, CRUK-funded trials, CASPS and PRIMETIME. CASPS is an international phase 2 trial of cediranib in the treatment of patients with alveolar soft part sarcoma.[12] The trial enrolled 47 patients across 12 sites in the UK, Spain and Australia and involved an internationally shipped Investigational Medicinal Product (IMP), completion and shipment of paper case report forms (CRFs), on site visits by the trial team for site initiation, monitoring visits and audit, additional hospital visits for patients for trial-specific assessments and provision of tissue samples. PRIMETIME is a postoperative avoidance of radiotherapy trial which recruited 1962 patients across 64 sites in the UK.[13] The trial used electronic data capture for collection of data, data linkage with NHS routine data sources and involved shipment of tissue samples for immunohistochemistry testing and subsequent long-term sample storage.

These two trials were selected because of their contrasting designs (International Clinical Trial of an Investigational Medicinal Product (CTIMP) vs national non-CTIMP), the trials were complete (meaning we could test the method on the entirety of a trial), and activity information was readily available.

Having identified the trial activities using the process map and guidance, activity data was gathered from the trial protocol, Site Investigator File, funding application, trial specific databases, systems and trackers, and through discussion with the trial teams. Once collated, the trial activity data were multiplied by the emission factors identified in the guidance to calculate a carbon footprint. Application to real life, complete trials resulted

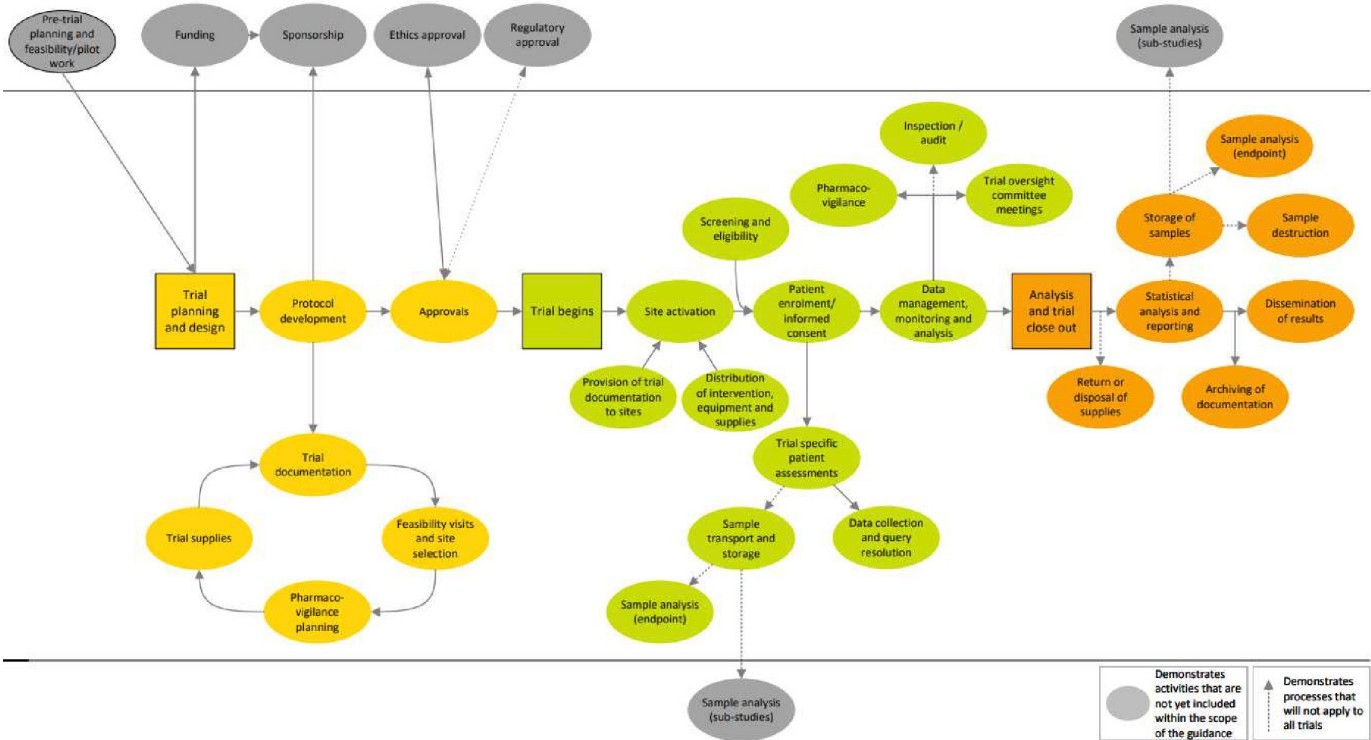

**Figure 1** A process map displaying the core activities of a clinical trial considered within the system boundary of the carbon footprinting calculations.

in iteration of the guidance and the addition of activities not previously identified, as well as their correlating emission factors for example, data linkage and copying images from trial-specific scans to CDs. The full calculations for CASPS and PRIMETIME can be found in online supplemental appendices C and D, respectively.

### Patient and public involvement
Patients and the public were not involved in this stage of the research to develop a method and guidance to carbon footprint a clinical trial. However, it is critical that patient partners are involved in this work now that a method is available, to seek their views on suggestions and recommendations to reduce the carbon footprint of clinical trials without reducing acceptability to participants.

### RESULTS
The guidance has been developed and used to calculate the carbon footprint of two clinical trials (results were presented at ICTMC in October 2022).

### CASPS trial
Applying the guidance, the carbon footprint of CASPS was estimated to be 72 tonnes (71 939 kgCO$_2$e). The carbon footprint of a typical UK citizen is 12.7 tonnes CO$_2$e per year, meaning CASPS has a footprint equivalent to around 6 UK citizens.[14] As shown below in figure 2, the largest contribution to the footprint was CTU emissions (23 488 kgCO$_2$e), followed by trial specific meetings and travel (20 465 kgCO$_2$e), largely due to the sites in Australia which required internal flights for site visits.

Despite the analysis of samples for a substudy being out of scope, central laboratory activities still contributed significantly to the footprint due to storage of samples in −80° freezers for up to 10 years.

### PRIMETIME trial
The total carbon footprint of PRIMETIME was calculated to be 89 tonnes (89 360 kgCO$_2$e). This is equivalent to seven times the annual carbon footprint of a UK citizen.

Figure 3 demonstrates that the largest contribution to the PRIMETIME carbon footprint was patient travel and assessments (60 823 kgCO$_2$e), followed by CTU emissions (11 733 kgCO$_2$e) and data collection and exchange (7572 kgCO$_2$e).

The PRIMETIME trial consisted of two treatment cohorts. In the standard of care cohort, the intervention comprised standard of care radiotherapy and annual follow-up mammograms were undertaken for 5 years to assess disease outcome. In the other group, no radiotherapy was given and patients received annual follow-up mammograms for 10 years to assess disease outcome. The additional five scans and associated patient travel were included in the PRIMETIME carbon footprint.

### Further testing
The current version of the detailed guidance is undergoing further iterative testing on a selection of trials from eight participating CTUs as described in table 1. Through its wider application, we hope to capture all possible clinical trial activities and emission factors across a wide range of trial designs. Although the

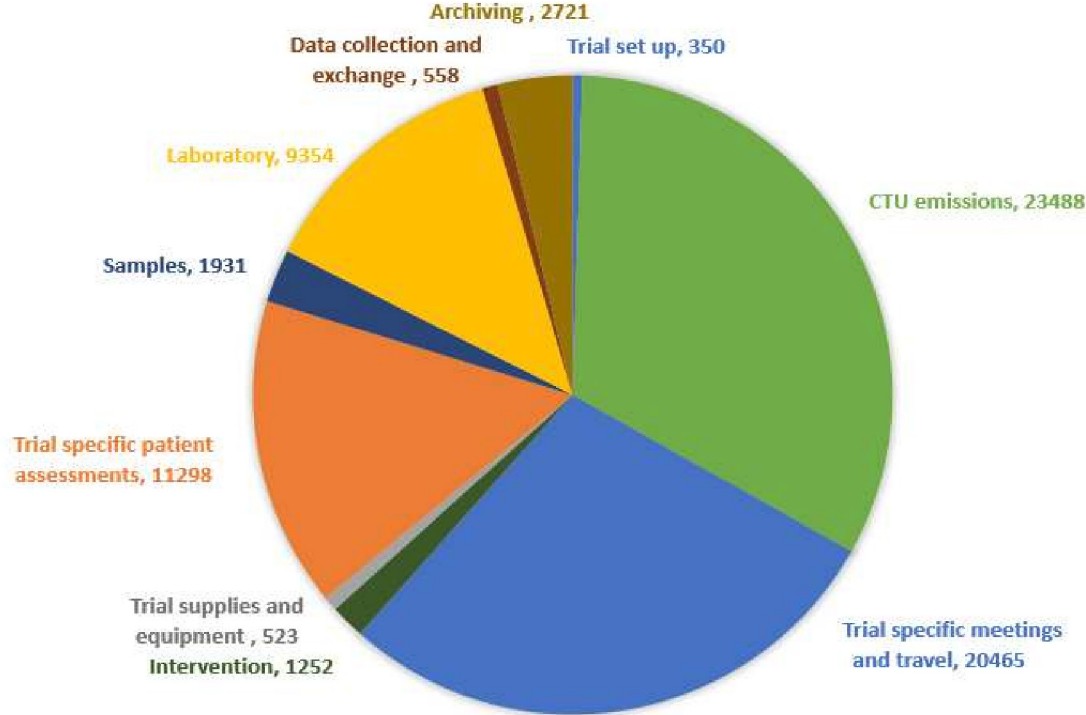

**Figure 2** A breakdown of the contributions to CASPS carbon footprint. CTU, clinical trials unit.

calculations required to carbon footprint the case studies in their entirety were extensive, through further testing and refinement of the guidance, we hope to identify hotspots so that future iterations of the guidance can be streamlined to enable data collection to be less resource intensive.

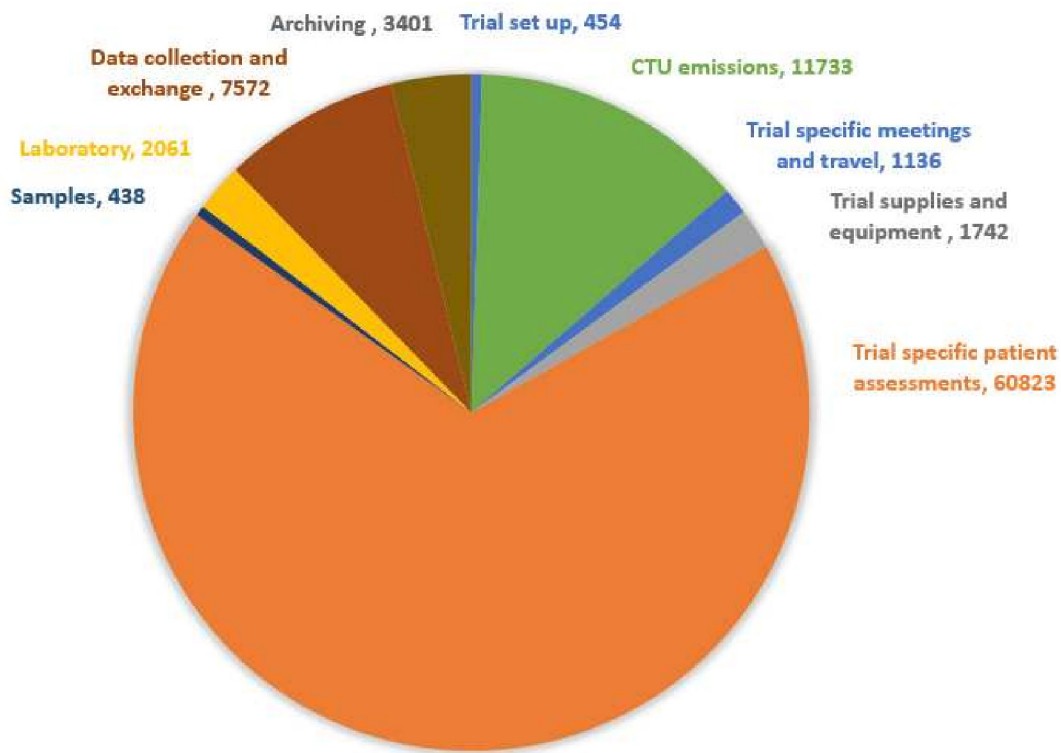

**Figure 3** A breakdown of the contributions to PRIMETIMEs carbon footprint. CTU, clinical trials unit.

The project team are also collecting qualitative data on the use and application of the guidance document. This will include how academic trialists managed this work alongside day-to-day trial management; the staff resource and time required to complete the calculations; any barriers to carrying out this work; how user friendly the guidance was and how much technical knowledge and support were required to complete the calculations.

## DISCUSSION

We have developed guidance that trialists could use to carbon footprint clinical trials and subsequently inform lower carbon trial design and delivery.

The detailed guidance document, enclosed as an appendix, describes a method that can be used to calculate the carbon footprint of publicly funded/investigator initiated, CTIMP and non-CTIMP (inc complex intervention) trials conducted in the NHS. Although the governance, funding and intent of commercially sponsored clinical trials can be different, the individual trial activities are comparable with publicly funded/investigator initiated research and as such this method could be used to footprint the activities within any clinical trial. However, the scope and assumptions defined in the guidance are tailored to academic, publicly funded clinical trials. The guidance is currently undergoing testing on a wider selection of trials managed by collaborating UK CRC Registered and international CTUs. The project team plan to publish the results of all additional trials footprinted as part of the subsequent testing phase and updates to the guidance this will inevitably generate, on completion of the currently funded project. The collaboration is open to, and welcomes new partners. Collaborators can receive support in using and applying the guidance as well as help sourcing new emission factors through webinars and drop in clinics facilitated by the MRC-NIHR TMRP Greener Trials Group. The Group aims to provide a forum for the development and dissemination of greener research practice.

It is important to note that this guidance accounts only for the carbon footprint of a clinical trial. It does not include other metrics that are important to consider in a full life cycle analysis when evaluating environmental impact and potential trade-offs, including water use, land use, waste and those relating to social and economic impacts. In addition, activities performed by the manufacturers of interventions used in the trial and activities performed by regulators to review and approve clinical trials have been excluded from the scope of the guidance, as these are not activities that trialists working on publicly funded and/or investigator initiated clinical trials can influence or control.

It is necessary to note that the validity of the carbon footprinting results the guidance generates is highly dependent on the quality and completeness of the activity data collated and the specific emission factors used. We recognise that more suitable or up to date emission factors

than those provided may be available or will become available as knowledge around the subject grows, but these could not be included as they are not publicly or freely available.

A number of assumptions are defined in the guidance to ensure consistency of application to different clinical trials and a degree of pragmatism applied, such as using average occupancies for members of hospital staff. All assumptions made are described in full in the guidance and can be interrogated in the real-life example calculations included in online supplemental appendices C and D.

Previous research clearly demonstrated that clinical trials have an impact and where the carbon hotspots might be (trial team commuting, energy use in research premises, trial travel and freight). It is difficult to compare the results of the case studies to those of the CRASH-1 and CRASH-2 carbon audits as they were large international trials which enrolled 10008 and 20211 patients, respectively, and emissions were calculated over a single, 1 year period.[15] We also grouped activities in different ways; in the CRASH audit the footprint of all deliveries was presented together, whereas here deliveries were calculated within their specific module, for example, delivery of site investigator files was included with 'trial set up' activities. Trial coordination centre emissions were the largest and second largest contributors in the CRASH trials, similarly in the case study results presented here, 'CTU emissions' was the largest contributor to the CASPS footprint and the second largest in PRIMETIME. Trial team commuting and energy use in research premises, which were calculated together in the current guidance under 'CTU emissions', also accounted for the most $CO_2e$ emissions in the 2009 study conducted by Lyle *et al*. This study calculated the average emissions of 12 UK trials to be 78.4 tonnes, comparable to the 72 and 89 tonne carbon footprints of CASPS and PRIMETIME. Patient travel was not calculated in the CRASH trials and was only the fourth largest contributor to emissions in the Lyle study. Conversely, in the case study results presented here 'trial specific patient assessments' was the largest contributor to PRIMETIME emissions and included both patient travel and procedures.

The results presented are in agreement with previously identified hotspots, however, the case studies have also identified different and new hotspots, such as data collection and exchange and laboratory associated activities. The results also show that the % contribution by activity varied across the two case studies, demonstrating the need to collect data from more trial types and designs to capture all possible hotspots.

As well as refinement and enhancement of the guidance to ensure it is user-friendly and can be applied to as many clinical trial types as possible, more trials must be footprinted to understand how to define the scope of what is considered a clinical trial and how best to apply the guidance.

The carbon footprint associated with the manufacture and delivery of the trial intervention are out of scope

in the current guidance. However, when implementing the results of a clinical trial it will also be important to consider the footprint of the recommended intervention. This latter point is exemplified by the PRIMETIME trial, where delivery of the trial has a notable carbon footprint. However, if the trial results recommend the avoidance of radiotherapy for some breast cancer patients, there would be a significant carbon saving because standard of care radiotherapy and associated patient travel would no longer be required (the carbon footprint associated with delivering 15 doses of standard of care radiotherapy is 94.19 kgCO$_2$e per patient, consisting of 7.19 kg CO$_2$e for the radiotherapy and 87 kgCO$_2$e for the travel required).[9 16] Further work is required to develop guidance on the interpretation of carbon footprinting results, so that the impact of conducting a clinical trial can be balanced against the improvement an intervention may make both in terms of patient outcomes and the carbon footprint of the intervention once incorporated into standard of care.

The results of the future work with collaborating CTUs summarised in table 1 will be published in due course and will include discussion and comparison of the hotspots, as well as evaluation of the inter-observer reliability and reproducibility. We will also look at the time commitment required to complete the calculations and how the guidance can be improved for use by those without specialist carbon footprinting knowledge. In the case studies the guidance was applied retrospectively to completed trials, however, ultimately the aim is to use the guidance prospectively at the trial design stage. We anticipate application in this scenario will be less time and resource intensive, but this will be analysed and discussed in the next paper, as we plan to include trials which are currently in development. Our findings will contribute to ongoing work coordinated by the SHC to develop an online, open-access carbon footprint calculator in collaboration with a group of pharmaceutical industry representatives conducting similar work. Through this collaboration, the generalisability of a common method and the scope of its application to academia and industry will be examined.

There remains much to do to help shape the future of sustainable clinical trials and support the move towards more responsible research. Investigation of carbon trade off decisions with patient and public involvement is necessary to enable recommendations to be made to reduce the carbon footprint of clinical trials, without reducing the clinical value of the trial and acceptability to participating patients and work with patient partners to shape the next steps of this important research is planned. Dissemination and engagement activities, including training and practical support for research teams incorporating carbon calculations into routine trial design, will be required to help drive the paradigm shift to responsible greener research.

As we gather more data on the estimated carbon footprint of planned clinical trials, further consideration should be given to how various stakeholders are going to use and consider the information generated. Funding panels often refer to cost per patient, a concept proposed for this area in terms of the carbon emission per patient. Care would be needed to interpret this statistic; as is the case with the cost per patient, carbon footprint per patient may discriminate against rare disease trials with a high health burden, as it does not include information on the nature and consequences of the condition under study. Further consideration is also required to determine a meaningful way to present the carbon footprint information and provide context to the public. A suggestion has been made that graphical representation in terms of the equivalent number of UK citizens' footprints could be more easily understood, but further research is needed.

Ultimately, a method and associated guidance describing how to carbon footprint a clinical trial will support various organisations including universities, health service providers and research funders to meet their carbon reduction commitments and allow stakeholders to incentivise a reduction of the carbon footprint of trials in their portfolios. We have produced this guidance as the first step of a strategy to help towards achieving this aim.

**Acknowledgements** Grateful thanks to Environmental Resources Management for their technical support and expertise in the development of the method and sourcing some emission factors.

**Collaborators** Low Carbon Clinical Trials Group Co-authors: Fiona Adshead (Sustainable Healthcare Coalition, UK); Rustam Al-Shahi Salman (Edinburgh Clinical Trials Unit, University of Edinburgh, UK); Craig Anderson (The George Institute, Australia); Emma Bedson (Liverpool Clinical Trials Centre, UK); Judith Bliss (Clinical Trials and Statistics Unit, The Institute of Cancer Research London, UK); Ana Boshoff (Imperial Clinical Trials Unit, Imperial College London, UK); Xiaoying Chen (The George Institute, Australia); Denise Cranley (Edinburgh Clinical Trials Unit, University of Edinburgh, UK); Peter Doran (University of Galway, Ireland; Carrol Gamble, Liverpool Clinical Trials Centre, UK); Kerenza Hood (Centre for Trials Research, Cardiff University, UK); Naomi McGregor (Newcastle Clinical Trials Unit, Newcastle University, UK); Carolyn McNamara (Technology & Innovation, KCR, UK); Elis Midha (School of Medicine, Cardiff University, UK); Keith Moore (Sustainable Healthcare Coalition, UK); Alexis M Perkins (Imperial Clinical Trials Unit, Imperial College London, UK); Sarah Pett (MRC Clinical Trials Unit at UCL, Institute of Clinical Trials and Methodology, UCL, London, UK); Matthew R Sydes (MRC Clinical Trials Unit at UCL, Institute of Clinical Trials and Methodology, UCL, London, UK).

**Contributors** JG made substantial contributions to the conception and design of the work, acquired, analysed and interpreted date for the work, drafted the work, reviewed it critically for important intellectual content, gave final approval of the version to be published, ensured questions related to the accuracy or integrity of any part of the work were appropriately investigated and resolved. LF made substantial contributions to the conception and design of the work, acquired, analysed and interpreted date for the work, drafted the work, reviewed it critically for important intellectual content, gave final approval of the version to be published, ensured questions related to the accuracy or integrity of any part of the work were appropriately investigated and resolved. PRW made substantial contributions to the conception and design of the work, acquired, analysed and interpreted date for the work, drafted the work, reviewed it critically for important intellectual content, gave final approval of the version to be published, ensured questions related to the accuracy or integrity of any part of the work were appropriately investigated and resolved. The Low Carbon Clinical Trials group made substantial contributions to the conception and design of the work and/or acquired, analysed and interpreted date for the work, reviewed the work critically for important intellectual content, gave final approval of the version to be published and ensured questions related to the accuracy or integrity of any part of the work were appropriately investigated and resolved.

**Funding** This project is funded by the National Institute for Health Research (NIHR) CTU Support Funding scheme [Award ID: NIHR135419].

**Disclaimer** The views expressed are those of the author(s) and not necessarily those of the NIHR or the Department of Health and Social Care.

**Competing interests** JG, LF, JB, PRW and CG have received NIHR funding to undertake this study (support for the manuscript). FA is the paid chair of the Sustainable Healthcare Coalition. KM is a member of the Sustainable Healthcare Coalition. RA-SS reports an NIHR Health Technology Assessment trial grant (CARE pilot trial) and an NIHR Efficacy and Mechanism Evaluation (EME) Programme Application Acceleration Award – International platform studies in precision medicine (CARE PREP) and consulting fees from Recursion Pharmaceuticals, McMaster University Population Health Research Institute, and Bioxodes, all paid to The University of Edinburgh. MRS reports grants from Astellas, Clovis Oncology, Janssen, Novartis, Pfizer, and Sanofi-Aventis; consulting fees from Eli Lilly; speaker's fees from Lilly Oncology, Janssen and payment from Eisai, all unrelated to the topic of this Comment. SP reports grants/contracts from EDCTP; MRC; Gilead Sciences; Janssen-Cilag; ViiV Healthcare; NIHR; University of Minnesota and is a member of the TIPAL DSMB. PD reports grants from Health Research Board of Ireland and Enterprise Ireland for Peer Reviewed research funding. CSA is the Vice President of the World Stroke Organisation and is Editor-in-chief of Cerebrovascular Diseases, and holds Research grants from National Health and Medical Research Council (NHMRC) of Australia, Medical Research Council (MRC) of the UK, Takeda and Penumbra (paid to the George Institute).

**Ethics approval** Not applicable.

**Provenance and peer review** Not commissioned; externally peer reviewed.

**ORCID iD**
Jessica Griffiths http://orcid.org/0009-0006-2406-3796

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
