## [Reviewer comments · BMJ Open]

ARTICLE DETAILS

TITLE (PROVISIONAL)	Quantifying the carbon footprint of clinical trials: guidance development and case studies
AUTHORS	Griffiths, Jessica; Fox, Lisa; Williamson, Paula; Low Carbon Clinical Trials Group, group authorship

VERSION 1 – REVIEW

REVIEWER	Trelle, Sven University of Bern, CTU Bern
REVIEW RETURNED	25-Jul-2023

GENERAL COMMENTS	Disclaimer: I am not a specialist in carbon footprint calculations. The authors aimed to develop a framework/methodology to quantify the carbon footprint of clinical trials. In the manuscript, they describe how they approached the problem. I think this is a very important and timely topic. I am not aware of any relevant studies looking into it comprehensively. Therefore, this report is of great interest and a welcome first step. However, I do not think that the manuscript describes "Original research" and I try to explain this below. I do not know the different article types at BMJ Open, which is why I recommend to reject the submission although I do think the project deserves publication in a scientific journal! But I also think that the current description is too superficial. Generally: The title says "... guidance development and pilot study". I think a lot of details are missing to allow readers to follow the guidance was developed and scrutinize it. I also think that the approach is at least partly unsystematic. This is per se not a problem if it is clearly described as such. But this in turn means, that the described project is not scientific research. I am not sure whether the two case studies constitute a pilot study: no aims of the pilot are described, no "endpoints" or measures specified, the sampling of the two trials is missing (why specifically these two?) and the results section describes only the carbon footprint and contributions but nothing about the application of the framework or similar. Therefore, these are rather case studies for me. Specific aspects: The approach relies on three main steps: 1. process map of all clinical trial activities 2. identification of trial-specific activities 3. applying a carbon footprint calculation to each activity Each of these is essential and therefore deserves detailed description in how they were developed/selected.
---

	The approach to process mapping is described as a rather unsystematic exercise: why was only the NIHR Clinical Trials Toolkit used? Maybe the processes only apply to NIHR CTUs (which would be a pity ...)? If so, the article should clearly describe this very narrow application explicitly (in title and text). The assumptions in the appendix actually mentions this but I do not think it is appropriate to have this major limitation in the appendix only somehow (it does not make explicit that the framework does not necessarily apply to commercial clinical research). Which CTUs participated in the meetings, which functions/roles were present, how where the meetings done (informally or a formalized process) etc.? Was there any 'validation' done of the mapping? Important activities are not covered. This might be okay in principle but justification is missing. Further, some of the activities that are not covered are actually very important. For example, leaving out manufacturing of the intervention (IMP or MD) implies that this is not a trial-specific activity but this is not true for a lot of trials (especially early phase i.e., before approval), why is the approval process not covered as this is truly trial-specific, and finally, information flow for meetings might be minor but appear not too complicated to quantify. It remains unclear how the footprint calculator was selected. Was there any systematic search and selection process? The only criterion appears to be that the calculator had to be freely available. This is understandable but not a very strong argument. I think sensitivity analyses with other calculators (plural on purpose) would be essential (if not mandatory) even though they might not be freely available. The detailed guidance shows print screens from MS Excel or at least some calculator. It would be important and helpful to provide this also as an appendix. As mentioned above, if the project was rather unsystematic, it is still of interest to be reported. However, this should be done in an appropriate format e.g., as a commentary. If the approach was systematic, much more details are needed.
--	--

VERSION 1 – AUTHOR RESPONSE

Comments from Reviewer 1

Comment: I think this is a very important and timely topic. I am not aware of any relevant studies looking into it comprehensively. Therefore, this report is of great interest and a welcome first step.
Response: Thank you for the positive feedback.

Comment: Manuscript does not describe "Original research", is at least partly unsystematic and is not scientific research (paraphrased).
Response: We accept the Editor's proposal to resubmit as a Communication article.

Comment: I am not sure whether the two case studies constitute a pilot study: no aims of the pilot are described, no "endpoints" or measures specified, the sampling of the two trials is missing (why specifically these two?) and the results section describes only the carbon footprint and contributions but nothing about the application of the framework or similar. Therefore, these are rather case studies for me.

Response: We did not conduct a pilot study, rather we 'piloted' the method on two trials which were chosen specifically to see if the list of activities we had assembled would apply and be sufficient. We have amended the title and wording throughout so that the trials are now referred to as 'case studies' rather than a pilot study. In terms of application of the framework, we noted in the original text that application of the guidance to these case study trials resulted in identification of activities not yet included in the guidance, and the guidance was therefore refined as a result of footprinting the case study trials.

Comment: The approach relies on three main steps: 1. process map of all clinical trial activities 2. identification of trial-specific activities 3. applying a carbon footprint calculation to each activity. Each of these is essential and therefore deserves detailed description in how they were developed/selected.

Response: The description of the three steps in the method have been expanded in line with responses to subsequent reviewer comments around the process map development and emission factors.

Comment: Why was only the NIHR Clinical Trials Toolkit used? Maybe the processes only apply to NIHR CTUs? If so, the article should clearly describe this very narrow application explicitly (in title and text). The assumptions in the appendix actually mentions this but I do not think it is appropriate to have this major limitation in the appendix only somehow (it does not make explicit that the framework does not necessarily apply to commercial clinical research).

Response: The processes do not only apply to NIHR CTUs, they could be applied to any theoretical clinical trial, including commercial clinical research. However, the intended audience of the guidance is Clinical Trials Units (CTUs) and trialists designing and delivering publicly funded/investigator initiated clinical research. We assembled the process map using the suite of Trial Management SOPs in the ICR-CTSU Quality Manual and information gathered from participating CTUs with reference to the NIHR Toolkit to try and ensure we had identified as many clinical trial activities as possible. Although the governance, funding and intent of commercially sponsored research is different, trial activities are comparable with publicly funded/investigator initiated research and as such this guidance could be used to footprint the activities within any clinical trial. However, the scope and assumptions within the guidance are specific to academic, publicly funded clinical trials. We have added a clarification regarding application of the guidance to commercial research to paragraph two of the discussion and have also tried to make this clearer in the text throughout.

Comment: Which CTUs participated in the meetings, which functions/roles were present, how where the meetings done (informally or a formalized process) etc.? Was there any 'validation' done of the mapping?

Response: CTU 'discovery meetings' were held with five CTUs – Cardiff Centre for Trials Research, Liverpool Clinical Trials Centre, Imperial Clinical Trials Unit, Edinburgh Clinical Trials Unit and MRC Clinical Trials Unit at UCL. Meetings were conducted informally with the purpose of discussing the CTU trial portfolio to identify clinical trials and trial activities not yet represented in the process map. Discussions resulted in refinement of the process map and identification of the trial within the CTU to footprint which would result in the inclusion of the widest variety of trial types and activities in the test phase. The process map was not formally validated, though informally reviewed by the project team. The project team felt that the process map did not require formal validation, as it does not need to be specific and exhaustive in terms of the activities included in a clinical trial. The purpose of the map is to give a high-level summary of trial processes, so teams are prompted to consider the activities occurring at each step of the trial.

Comment: Important activities are not covered. This might be okay in principle but justification is missing. E.g. manufacturing of the intervention (IMP or MD) implies that this is not a trial-specific activity

Response: We agree that it would be incorrect to consider that IMP manufacture is not a trial-specific activity. However, it is not an activity that trialists working on publicly funded clinical trials can influence or control. The intention of the guidance is to be the first step to trialists having information that they can use to inform lower carbon trial design. In order to include IMP emissions, trialists would need to work with the manufacturer and request emission factors or life cycle data available, currently there is no requirement for manufacturers to provide this information or guidance on how they should produce it consistently. Therefore this would potentially be a time consuming exercise that may not result in reliable data. The authors are working with the NHS Greener Trials team and pharmaceutical partners via the industry Low Carbon Clinical Trials group to understand how best to incorporate information on the footprint of clinical interventions, but until more data on this is publicly available it is not possible to include this in the scope of this guidance. We have added a sentence to clarify the reason for exclusion of IMP/MD manufacture in paragraph 3 of the discussion.

Comment: Why is the approval process not covered as this is truly trial-specific?

Response: We agree that the approval process is trial specific. The CTU staff time allocated to a trial is included in the CTU emissions, as is generation of all trial-specific essential documentation. However, the time spent on the approval process by the REC, HRA, and MHRA is not an activity that trialists working on publicly funded clinical trials can influence or control and this aspect has therefore been left out of this first iteration of the guidance. The co-authors plan future work with all clinical trial stakeholders, including regulators to better understand their contribution to the footprint of the trial. We have added a sentence to clarify the reason for exclusion of regulatory approvals in paragraph 3 of the discussion.

Comment: Information flow for meetings might be minor but appear not too complicated to quantify.

Response: We agree and in fact inclusion of this assumption is an error as all information flow relating to a trial, e.g. travel to meetings, emails and documentation, has been included in the scope and were calculated in both case studies. We have deleted this assumption from the guidance document in the supplementary materials.

Comment: It remains unclear how the footprint calculator was selected. Was there any systematic search and selection process? The only criterion appears to be that the calculator had to be freely available. This is understandable but not a very strong argument. I think sensitivity analyses with other calculators (plural on purpose) would be essential (if not mandatory) even though they might not be freely available.

Response: We did not use generic carbon calculators to complete any of the calculations. The main sources of emission factors used were GHG conversion factors, Ecoinvent and data from the SHC care pathway carbon calculator. Where factors were not available in these publicly available sources, peer-reviewed, published papers, articles and product specifications were used e.g. for pathology tests and electronic devices. Environmental Research Management (ERM) acted in an advisory role on the project and provided technical advice on the emissions factors chosen for use in the guidance. This is an evolving field in which emission factors are updated and created frequently with increasing industry specific research. It is correct that the choice of emission factor has a bearing on the final carbon footprint, and we have described that as a limitation, however our intention is that the guidance be used to compare trial activities to inform lower carbon trial design, rather than produce an absolute carbon footprint. Therefore, the current priority is to provide trialists with a method so they may start to consider the carbon footprint of trials they design and be able to compare alternative approaches that could reduce their carbon footprint. We also wanted to ensure the guidance developed could be used without the need for purchase of licensed emission factor data which would be a barrier for publicly funded trialists.

Comment: The detailed guidance shows print screens from MS Excel or at least some calculator. It would be important and helpful to provide this also as an appendix.

Response: The excel spread sheet is a list of publicly available 2021 GHG conversion factors that ERM collated and shared for use. We have included the section required for the specific activity listed, but it is not possible and would not be relevant to include the whole document as there are other emission factors included which are not relevant to clinical trials.

Comment: If the project was rather unsystematic, it is still of interest to be reported. However, this should be done in an appropriate format e.g., as a commentary. If the approach was systematic, much more details are needed.

Response: We accept the Editor's proposal to change this to a Communication article.